# Polydopamine-Assisted Rapid One-Step Immobilization of L-Arginine in Capillary as Immobilized Chiral Ligands for Enantioseparation of Dansyl Amino Acids by Chiral Ligand Exchange Capillary Electrochromatography

**DOI:** 10.3390/molecules26061800

**Published:** 2021-03-23

**Authors:** Yuanqi Gui, Baian Ji, Gaoyi Yi, Xiuju Li, Kailian Zhang, Qifeng Fu

**Affiliations:** 1School of Pharmacy, Southwest Medical University, Luzhou 646000, China; 20200599120019@stu.swmu.edu.cn (Y.G.); jibaian1997@163.com (B.J.); yigaoyi169216@163.com (G.Y.); zkl66@swmu.edu.cn (K.Z.); 2School of Pharmacy, Tongren Polytechnic College, Tongren 554300, China

**Keywords:** chiral ligand exchange capillary electrochromatography, enantioseparation, L-arginine, polydopamine, enzyme kinetics study

## Abstract

Herein, a novel L-arginine (L-Arg)-modified polydopamine (PDA)-coated capillary (PDA/L-Arg@capillary) was firstly fabricated via the basic amino-acid-induced PDA co-deposition strategy and employed to constitute a new chiral ligand exchange capillary electrochromatography (CLE-CEC) method for the high-performance enantioseparation of D,L-amino acids (D,L-AAs) with L-Arg as the immobilized chiral ligand coordinating with the central metal ion Zn(II) as running buffer. Assisted by hydrothermal treatment, the robust immobilization of L-Arg on the capillary inner wall could be facilely achieved within 1 h, prominently improving the synthesis efficiency and simplifying the preparation procedure. The successful preparation of PDA/L-Arg coatings in the capillary was systematically characterized and confirmed using several methods. In comparison with bare and PDA-functionalized capillaries, the enantioseparation capability of the presented CLE-CEC system was significantly enhanced. Eight D,L-AAs were completely separated and three pairs were partially separated under the optimal conditions. The prepared PDA/L-Arg@capillary showed good repeatability and stability. The potential mechanism of the greatly enhanced enantioseparation performance obtained by PDA/L-Arg@capillary was also explored. Moreover, the proposed method was further utilized for studying the enzyme kinetics of L-glutamic dehydrogenase, exhibiting its promising prospects in enzyme assays and other related applications.

## 1. Introduction

Chirality is one of the crucially fundamental characteristics of nature [1]. A number of compounds which compose the building blocks of life are enantiotropic chemicals, and the different stereostructures and optical rotation performance between the enantiomers have major effects on the metabolism and normal physiological activities of the life systems [2,3]. Amino acids (AAs) are ubiquitous in all forms of life, and they are basic components of proteins and act as signal transmitters [4,5]. Moreover, free AAs also play a crucial part in numerous life activities [6,7]. However, there are striking differences in vital movement between D-AAs and L-AAs. L-AAs have been considered essential in humans, whereas D-AAs are closely correlated with the nervous and endocrine systems [8,9,10]. Therefore, the question of how to obtain highly optically pure amino acid enantiomers has long been of great significance, especially for the food and pharmaceutical industries.

Among all the methods for chiral analysis of AAs, chiral ligand exchange capillary electrophoresis (CLE-CE) has recently received tremendous attention in chiral separation of AAs owing to its benefits of short analysis time, high convenience, and controllable enantiomer migration order [11,12,13,14]. The chiral recognition mechanism of CLE-CE is based on the formation of diastereomeric ternary metal complexes between the chiral ligands and the analytes [15]. More specifically, chiral separation can be realized owing to the different stability constants of the metal complexes, as shown in the following equations:(1)$$M{\left(L-Sel\right)}_{n}+L-A⥦M{\left(L-Sel\right)}_{n-1}\left(L-A\right)+L-Sel$$
(2)$$M{\left(L-Sel\right)}_{n}+D-A⥦M{\left(L-Sel\right)}_{n-1}\left(D-A\right)+L-Sel$$
where *M* is the central metal ion; *Sel* is the chiral selector; *A* is the analyte. Although CLE-CE has been reported by many academic research studies, it still faces the pressing challenges of limited chiral ligands, unsatisfactory enantioseparation performance, and narrow application range [16,17]. In order to address the above issues, two main strategies could be applied. The first is to develop some novel chiral ligands, including AA derivatives, polymer-based chiral ligands, and AA ionic liquids [18,19,20]. Although good enantioseparation performance of some D,L-AAs can be realized, many of these new ligands are not yet commercially available, and their synthesis processes are also sophisticated and time-consuming. The other alternative is to construct unique CLE capillary electrochromatography (CLE-CEC) systems. The chiral separation mechanism of CLE-CEC is also based on the different stability constants of the ternary metal complexes between the chiral ligands and the analytes. Slightly different from CLE-CE, in CLE-CEC, the chiral ligands can be immobilized on the capillary’s inner surface or filled in the column as the free chiral ligand. Whereafter, the D,L-AAs can be separated, benefitting from the strong synergistic effect between the immobilized and free ligands. Therefore, CLE-CEC can effectively combine the great enantioseparation efficiency of CLE-CE and the high recognition of stationary phases, which has aroused much interest [21,22,23,24,25,26]. For instance, Qi and coworkers synthesized an L-arginine methyl ester-derived block copolymer and applied it as the immobilized ligands coordinated with Zn(II) and free ligands for enantioseparation of D,L-AAs [9]. Nevertheless, the previously reported CLE-CEC methods encounter certain defects, such as the time-consuming, sophisticated fabrication processes and the overuse of organic solvents in the buffer solution, which has negative effects on biological sample analysis. Therefore, exploiting the facile and rapid strategies to develop an environmentally friendly CLE-CEC system without adding organic solvents for chiral separation is still under urgent demand but challenging.

Mussel-inspired polydopamine (PDA)-derived coatings have attracted great attention in the field of CEC because they could be utilized as the strong linker to immobilize various functional modifiers [27,28,29]. Owing to this advantage, a few chiral selectors, such as β-cyclodextrin, protein, and DNA, have been immobilized to the inner surface of the capillary for electrochromatographic enantioseparation through a PDA-assisted modification method [30,31,32]. However, according to current knowledge, the possible applications of the PDA-assisted modification strategy to immobilize chiral ligands for high-efficiency chiral separation of D,L-AAs with CLE-CEC have not been studied so far.

In previous studies, our group prepared a novel octadecylamine (ODA)-modified PDA-coated capillary through an organic amine-triggered PDA-coating strategy [33,34].

In these cases, ODA was used to induce dopamine polymerization; meanwhile, it was reacted with dopamine and co-deposited into a PDA coating. Inspired by the prior works, we explored the possibility of PDA-assisted immobilization of L-arginine (L-Arg) on the capillary inner wall for establishing an innovative open-tubular CLE-CEC system. Specifically, L-Arg, a basic amino acid with plentiful amino groups, may also be utilized simultaneously as the trigger for the growth of PDA and the chiral ligand modified in the PDA coating via the strong coupling between amine and catechol groups.

In this work, a novel L-Arg-modified PDA-coated capillary (PDA/L-Arg@capillary) was firstly prepared through the basic amino-acid-induced PDA co-deposition strategy and employed to constitute a new CLE-CEC system for the chiral separation of D,L-AAs with L-Arg as the immobilized ligand coordinating with Zn(II) in running buffer. Assisted by the hydrothermal treatment, the immobilization efficiency of L-Arg in the PDA coating was drastically increased and the facile and robust co-deposition process could be achieved within 1 h. Figure 1 presents a schematic of the formation of the PDA/L-Arg@capillary. The chemical components and morphologies of the resulting PDA/L-Arg layer were investigated and confirmed with a series of measures. By using this developed PDA/L-Arg@capillary, high-efficiency enantioseparation of D,L-AAs was easily achieved in the absence of organic solvents in buffer solutions. The excellent repeatability and stability of PDA/L-Arg@capillary were also verified. Moreover, the probable enantioseparation mechanisms of this new CLE-CEC system were investigated. Ultimately, the developed method was further utilized to study the enzyme kinetics of L-glutamic dehydrogenase (L-GLDH), exhibiting its promising prospects in enzyme assays and other related applications.

## 2. Results and Discussion

### 2.1. Characterization of L-Arg/PDA Coating

#### 2.1.1. Field Emission Scanning Electron Microscopy (FESEM)

The inner surface images of the bare capillary and PDA/L-Arg@capillary were characterized by FESEM, respectively. As demonstrated in Figure 2, the inner wall of the bare column is flat and featureless. After hydrothermal treatment with the mixed solution of L-Arg and dopamine, a rough inner surface with some visible small aggregates could be observed, which demonstrated the successful fabrication of an L-Arg modified PDA coating on the capillary inner wall.

#### 2.1.2. Fourier-Transform Infrared Spectroscopy (FTIR) and Attenuated Total Reflectance FTIR (ATR-FTIR)

FTIR and ATR-FTIR measurements were carried out to further confirm the formation of the L-Arg modified PDA coating. Figure 3a shows the FTIR spectra of dopamine, PDA, and PDA/L-Arg composite formed in the solution, which are accompanied by the deposition of PDA and PDA/L-Arg coatings. All three samples exhibited typical absorption of aromatic C=C at around 1650, 1542 cm^−1^, catechol –OH at 3370 cm^−1^, and amino groups at 3248 cm^−1^ [35]. On the other hand, the dopamine’s characteristic absorption peaks at 1326 cm^−1^ (C-H bending vibration), 1370 cm^−1^ (C-O-H deformation vibration), and 1239 cm^−1^ (C-N stretching vibration) were not discernible in the FTIR spectra of PDA and PDA/L-Arg composite, confirming that the PDA and PDA/L-Arg hybrid samples were free of dopamine and successfully synthesized [36,37,38]. In addition, the PDA/L-Arg hybrid sample not only retained the characteristic absorption features of PDA but also exhibited a new absorption peak at 1697 cm^−1^, which could be ascribed to the vibration of C=O in L-Arg molecules [39]. In addition, quartz plates whose ingredient was the same as the fused-silica capillary were coated with PDA/L-Arg coatings using the same steps as the PDA/L-Arg@capillary to further validate the co-deposition of PDA/L-Arg. ATR-FTIR measurements were employed to probe the possible PDA/L-Arg coatings on the quartz plates. Figure 3b illustrates the ATR-FTIR spectra of the unmodified plate and PDA-functionalized and PDA/L-Arg-functionalized plates. The characteristic absorption of 801 cm^−1^ (Si–OH) and 982 cm^−1^ (Si–O–Si) derived from silica were present for all three samples [40]. In comparison with the unmodified plate, PDA- and PDA/L-Arg-coated plates exhibited clear characteristic bands associated with PDA at around 1448 cm^−^^1^, 1524 cm^−1^, 1610 cm^−1^, 3211 cm^−1^, and 3395 cm^−1^, in accordance with the characterization results of FTIR. Moreover, the PDA/L-Arg-coated plate also exhibited an absorption feature of C=O vibration at 1692 cm^−1^, again confirming the successful modification of L-Arg in the PDA coating.

#### 2.1.3. Electroosmotic Flow (EOF)

The EOF mobility of the coated capillary could also be considered as persuasive proof of the functionalization of PDA/L-Arg. As shown in Figure 3c, the EOF mobilities of all capillaries rose gradually with the increasing pH value from 4.0 to 8.0. The highest EOF values were acquired on the bare capillary owing to the ionization of the surface silanol groups [33]. After the PDA layer was adhered to the capillary inner surface, the EOF values of the PDA-adhered capillary were remarkedly inferior to those of the unmodified capillary in the full pH range due to the presence of groups with positive charges on the PDA layer. As for the PDA/L-Arg@capillary, compared with the PDA-coated column, its EOF value was further reduced due to masking of the silanol groups and the presence of more N–H groups on the PDA/L-Arg layer [41].

### 2.2. Optimization of PDA/L-Arg@Capillary Preparation

#### 2.2.1. Effects of the Molar Ratio of Dopamine and L-Arg on Enantioseparation Capacity

First, 100 mM (20 mg/mL) dopamine was chosen as an appropriate concentration for fabricating PDA/L-Arg@capillary because of the optimal enantioseparation effect (Appendix A). Next, to further ascertain the optimal molar ratio of dopamine to L-Arg in the precursor solution, the effects of different concentrations of L-Arg (50–350 mM) in the presence of 100 mM dopamine on the enantioseparation of the model chiral analyte dansyl D,L-alanine (Dns-D,L-Ala) were intensively investigated. As shown in Appendix A, with the molar ratio of dopamine to L-Arg increased from 2:1 to 2:5, both the resolutions and migration times of Dns-D,L-Ala gradually increased on the PDA/L-Arg@capillary, which could be attributed to more L-Arg molecules existing in the hybrid coatings and increased interaction strength between PDA/L-Arg coatings and the model analyte. However, the enantioseparation efficiency began to decrease when the molar ratio of dopamine to L-Arg was further increased to 2:7, which might be because the excessively high alkalinity of the reaction solution resulted from the excessive L-Arg, which was not conducive to the self-polymerization of dopamine and the copolymerization between dopamine and L-Arg [42]. Therefore, 2:5 of dopamine to L-Arg was chosen as the optimal molar ratio for fabricating PDA/L-Arg@capillary in the next studies.

#### 2.2.2. Influences of Hydrothermal Temperature and Time of PDA/L-Arg Layer on Enantioseparation Capacity

The effect of hydrothermal reaction temperature on enantioseparation performance is illustrated in Appendix A. As hydrothermal temperature increased (90–120 °C), the migration times and resolution of Dns-D,L-Ala gradually increased, which indicated that the polymerization and deposition rate of PDA greatly increased and more and more L-Arg molecules were immobilized in the hybrid coating via Michael addition and Schiff-base reactions between PDA and L-Arg molecules. When the temperature further increased up to 160 °C, the enantioseparation performance decreased gradually, which might be because the excessive reaction temperature led to the partial carbonization and detachment of the PDA/L-Arg coating (Appendix A). We also investigated the influence of reaction time with the fixed hydrothermal reaction temperature at 120 °C. As revealed by Appendix A, the maximized resolution and theoretical plate numbers of the model analyte can be observed when the reaction time was up to 1 h. As the reaction time reached 1.5 h, the chiral separation efficiency decreased dramatically. To summarize, the PDA/L-Arg@capillary modified using the hydrothermal method at 120 °C for 1 h was selected in the following studies.

### 2.3. CLE-CEC Operation Conditions Optimization

The components and properties of the buffer solution, such as buffer pH, central ion concentration, and the ratio of Zn(II) to L-Arg, play key roles in the enantioseparation performance of PDA/L-Arg@capillary. Three pairs of Dns-D,L-AAs, including Dns-D,L-Ala, Dns-D,L-aspartate (Dns-D,L-Asp), and Dns-D,L-serine (Dns-D,L-Ser), were chosen as the model compounds. The effect of buffer pH ranging from 7.6 to 8.4 on the separation performance is illustrated in Figure 4a. The recognition of the tested compounds continuously increased as the pH values increased from 7.6 to 8.0. As the pH value of the buffer was greater than 8.0, the resolution of the three analytes began to decrease. On the other hand, the influences of Zn(II) ion concentrations in the running buffer were also examined in the range of 1.0–8.0 mM. As revealed by Figure 4b, while the concentration ratio of Zn(II) to L-Arg was fixed at 1:1, the best separation effect was obtained with the usage of 6.0 mM Zn(II) in the running buffer. Furthermore, Figure 4c depicts the results of varying the concentration ratio of Zn(II) to L-Arg from 3:1 to 3:5. The resolution continuously increased with the concentration ratio of Zn(II)/L-Arg from 3:1 to 1:1. Further increase in the concentration ratio would lead to the deterioration of enantioseparation effects. Furthermore, the influence of the applied voltage on enantioseparation was also studied (Appendix A). Taking account of the shortest retention time and best resolution, the operating voltage was set at −20 kV for further analysis. Thus, the CLE-CEC enantioseparation experiments were conducted by using an applied voltage of −20 kV and a buffer solution at pH 8.0 containing 6.0 mM L-Arg and Zn(II).

### 2.4. Enantioseparation Performance of PDA/L-Arg@Capillary

In order to further demonstrate the excellent enantioseparation performance of the developed CLE-CEC system with immobilized L-Arg chiral ligands, except for the aforementioned three Dns-D,L-AAs, it was further used for the enantioseparation of another eight Dns-D,L-AAs with the optimal conditions. As shown in Appendix A, only Dns-D,L-Ser could be completely separated (Rs > 1.50) on the bare column among all eleven Dns-D,L-AAs. By contrast, complete separation of eight Dns-D,L-AAs and partial separation of three Dns-D,L-AAs were successfully obtained on the PDA/L-Arg@capillary when using the same electrophoresis conditions, demonstrating the superior enantioseparation ability of the presented CLE-CEC method. In addition, the developed new CLE-CEC method without using organic solvent in the mobile phase possesses a more preferable chiral recognition capability than other previously recorded CLE-CEC methods, further indicating the great potential of a PDA-assisted chiral ligand modification strategy for constructing a high-performance CLE-CEC system (Appendix A) [8,9,26,43,44].

### 2.5. Repeatability and Stability of PDA/L-Arg@Capillary

Under the optimal CLE-CEC operation conditions, the repeatability and stability of the PDA/L-Arg@capillary was evaluated based on the relative standard deviations (RSDs) of retention time of Dns-D,L-AAs. The RSDs of intra-day, inter-day, and column-to-column were all below 5.8% (Appendix A). Moreover, PDA/L-Arg@capillary could be performed for 80 consecutive injections without significant signs of degradation in the retention time and the enantioseparation performance (Appendix A). These results confirmed that the PDA/L-Arg@capillary had good repeatability and stability, proving its promising prospects for practical application.

### 2.6. Exploration of Enantioseparation Scheme in Presented CLE-CEC

Chiral separation can be realized because of the varied stability of diastereomeric ternary mixed central metal ion between the chiral ligand and the analytes [15]. As mentioned above, baseline separation of eight Dns-D,L-AAs could be realized on the presented CLE-CEC systems. In contrast, only Dns-D,L-Ser was baseline separated and two Dns-D,L-AAs even had no signs of separation on the bare column. Furthermore, the chiral recognition capability on the PDA-coated capillary was also investigated. As can be seen in Figure 5, the resolutions of the model analyte obtained on the PDA-coated capillary were only slightly higher than that of the bare column, which was much lower than that of the PDA/L-Arg@capillary. Therefore, these results showed that the mobility differences in the ternary complexes on the bare and PDA-coated capillary were so slight that satisfactory enantioseparation effects were quite difficult to achieve. On the other hand, when the PDA/L-Arg@capillary was utilized in the absence of the free L-Arg in the running buffer, its enantioseparation performance was also unsatisfactory. Therefore, as depicted in Figure 6, it can be deduced that the excellent enantioseparation performance of the developed CLE-CEC system benefited from the strong synergistic effect between the immobilized L-Arg in the PDA/L-Arg coating and the free ligands in the buffer solutions.

### 2.7. Quantitative Determination of D,L-Glu

Since L-glutamic acid (L-Glu) was adopted as the substrate of L-GLDH, the quantitative determination of Dns-D,L-Glu was conducted using the proposed method. As a result, the detection limits and quantification limits of Dns-D,L-Glu were 15 and 50 μg/mL, respectively. The other data, including the good linear relationship and wide linearity ranges of Dns-D, L -Glu, are summarized in Appendix A. These results demonstrated that the fabricated PDA/L-Arg@capillary had the potential for the enzyme kinetic study of L-GLDH.

### 2.8. Enzyme Kinetic Study of L-GLDH

L-GLDH is a key enzyme for amino acid metabolism and urea production. In order to further illustrate the application potential and validate the practicability of the CLE-CEC system, the enzyme kinetic study of L-GLDH was performed by determining the maximum rate (V_max_) and Michaelis–Menten’s constant (K_m_). Based on the changes in substrate peak area, the values of V_max_ and K_m_ were calculated to be 414.94 mM/min and 5.41 mM. The data were similar to previously reported values [45], confirming the reliability of the developed CLE-CEC method in enzyme kinetics study.

## 3. Materials and Methods

### 3.1. Materials and Chemicals

D,L-amino acid enantiomers, dansyl chloride (Dns-Cl) and lithium carbonate, were purchased from Macklin Reagent Co., Ltd. (Shanghai, China). Dopamine hydrochloride, L-Arg, and zinc sulfate (ZnSO_4_) were purchased from Adamas Reagent Co., Ltd. (Shanghai, China). L-GLDH was provided by YuanYe Bio-Technology Co., Ltd. (Shanghai, China). Ultra-pure water was purified through an AWL-1002-H water purification system (Aquapro International Company LLC., Chongqing, China).

### 3.2. Apparatus

All the CE separations were conducted on an Agilent 7100 3D CE system (Agilent Technologies, Waldbronn, Germany). The pH values of the buffer solutions were ad-justed by a FiveEasy PlusTM-FE28 (Mettler Toledo, Shanghai, China). FESEM images were recorded on SU-8020 (Hitachi, Japan). The chemical components on the surface of the quartz plate were detected with ATR-FTIR spectroscopy (Nicolet iS50, Thermo Scientific Inc., Madison, WI, USA).

### 3.3. Buffer Solution and Sample Solution Preparation

The CLE-CEC running buffer (pH 8.0, adjusted by Tris) was composed of 100 mM boric acid, 10 mM ammonium acetate, 6 mM L-Arg, and 6 mM ZnSO_4_. Standard sample solutions of 2.0 mg/mL D,L-AAs were prepared in 40 mM lithium carbonate buffer (adjusted to pH 9.5 with 0.1 M HCl), then diluted to the desired concentrations with lithium carbonate solution for further analysis. For derivatization, 1.0 mL Dns-Cl (1.5 mg/mL, dissolved in acetonitrile) and 1.0 mL D,L-AAs were dissolved directly in 1.0 mL lithium carbonate buffer. The mixture was gently shaken for 2 min and then allowed to react at room temperature for 30 min away from light. All solutions were filtrated with 0.45-μm pore size polyethersulfone membrane and degassed by sonication for 3 min before use.

### 3.4. CLE-CEC Procedures

Before being measured, the modified capillaries (50 µm i.d. × 35 cm length; effective length, 26.5 cm) were flushed with running buffer for 15 min, and each 3 min with distilled water and buffer solution between consecutive injections. All samples were introduced hydrodynamically by 35 mbar for 5 s and separated with an operating voltage of −20 kV. The wavelength for detection was 214 nm, and the temperature of CLE-CEC procedures was maintained at 25 °C.

### 3.5. Fabrication of PDA/L-Arg@Capillary

The PDA/L-Arg@capillaries used in this study were facilely fabricated through filling the bare column with a mixture of dopamine and L-Arg, and then hydrothermal-assisted reacting at high temperature for a period. The preparation process is shown in Figure 1. Briefly, prior to modification, the capillary was accustomed to flushing using methanol (30 min), 1 M NaOH (30 min), ultra-pure water (5 min), 1 M HCl (30 min), and ultra-pure water (5 min). Subsequently, the capillary was dried with nitrogen stream. The preconditioned capillary was pumped with a freshly prepared mixture consisting of 250 mM L-Arg and 20 mg/mL dopamine for 5 min, then placed into the Teflon-lined bomb. Afterwards, the sealed Teflon-lined bomb was placed in an oven at 120 °C for 1 h to in situ grow an L-Arg/PDA layer on the capillary inner wall. Finally, the obtained capillary was washed with water and named the PDA/L-Arg@capillary.

### 3.6. Enzyme Kinetics Study of L-GLDH

The kinetic constants of L-GLDH were calculated by the proposed method. Various concentrations of L-Glu (dissolved in 250 mM phosphate buffer, adjusted to pH at 8.5) as the substrates were incubated with L-GLDH for 5 min at 40 °C. Afterwards, the enzymatic reaction was ceased by heating in boiling water for 15 min, and then centrifuged for 15 min (at 10,000 rpm). Then, the supernatants were extracted and derived using Dns-Cl. For accurate calculation of substrate concentration, different concentrations of Dns-D,L-Glu solutions from 15 to 800 μg/mL were applied to CLE-CEC, and then the calibration curve was constructed.

## 4. Conclusions

In summary, a novel CLE-CEC method was firstly developed with L-Arg as the immobilized chiral ligand for D,L-AAs enantioseparation. The hydrothermal-assisted in-situ PDA-based co-deposition strategy for immobilizing L-Arg chiral ligands could effectively avoid the time-consuming and sophisticated fabrication processes of the previously reported modification method. Benefitting from the strong synergistic effect between the immobilized L-Arg in the PDA/L-Arg coating and the free ligands in the buffer solutions, complete separation of eight Dns-D,L-AAs and partial separation of three Dns-D,L-AAs were successfully achieved on the constructed CLE-CEC system without the use of organic solvent in the running buffer. The fabricated PDA/L-Arg@capillary exhibited satisfactory repeatability and stability. Moreover, the presented method could further be utilized for studying the enzyme kinetics of L-GLDH, exhibiting its promising prospects in enzyme assays and other related applications.

## Figures and Tables

**Figure 1 molecules-26-01800-f001:**
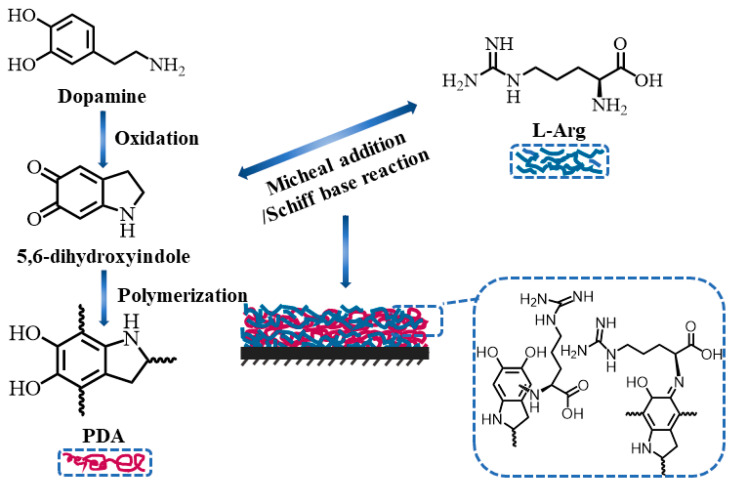
Schematic of the hydrothermal-assisted preparation of PDA/L-Arg@capillary.

**Figure 2 molecules-26-01800-f002:**
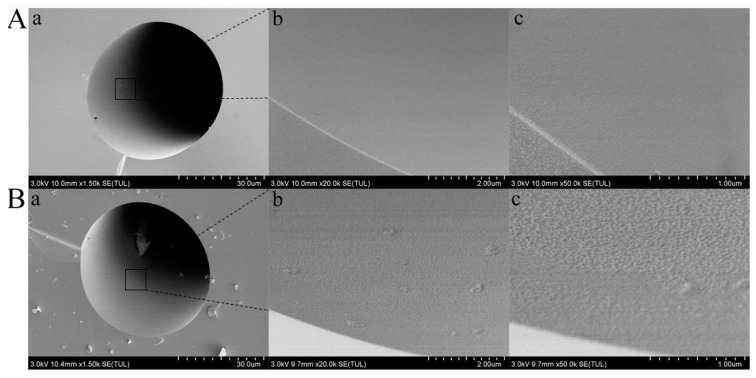
FESEM images of the inner wall of bare capillary (**A**) and PDA/L-Arg@capillary (**B**) with different magnification (a. 1.50 k, b. 20.0 k, c. 50.0 k).

**Figure 3 molecules-26-01800-f003:**
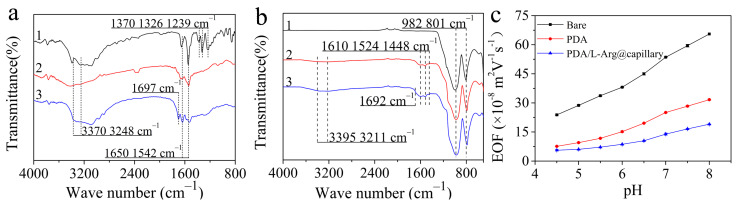
(**a**) FTIR spectra of dopamine (1), PDA (2), and PDA/L-Arg composite (3). (**b**) ATR-FTIR spectra of unmodified (1), PDA-coated (2), and PDA/L-Arg-coated quartz sheets (3). (**c**) The electroosmotic flow mobilities of bare column, PDA-coated column, and PDA/L-Arg@capillary at different buffer pH values ranging from 4.5 to 8.0. EOF marker, DMSO; Buffer, 100.0 mM boric acid, 10.0 mM ammonium acetate, 6.0 mM ZnSO_4_, 6.0 mM L-Arg; Capillary: 50 µm i.d. × 35 cm length (26.5 cm effective); sample injection pressure and time at 35 mbar and 5 s; 25 °C; voltage, −20 kV; UV detection at 214 nm.

**Figure 4 molecules-26-01800-f004:**
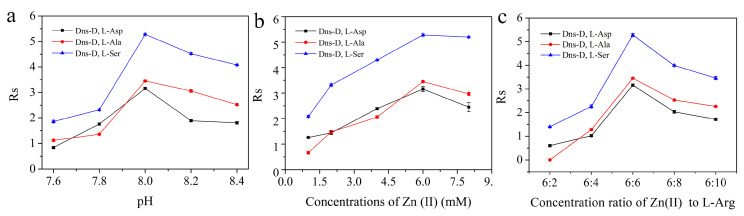
Effects of buffer pH (**a**), Zn(II) concentration (**b**), and the concentration ratio of Zn(II) to L-Arg (**c**) on the resolution of Dns-D,L-AAs. Chiral ligand exchange capillary electrochromatography conditions, 100.0 mM boric acid, 10.0 mM ammonium acetate at different pH and concentrations of Zn(II) and L-Arg. Other conditions are the same as Figure 3.

**Figure 5 molecules-26-01800-f005:**
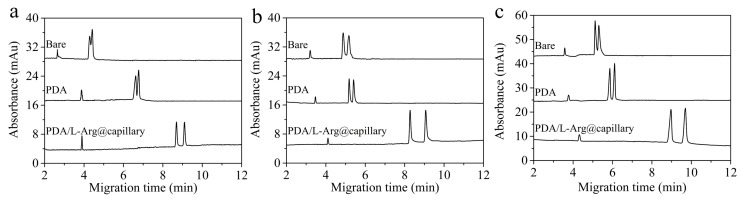
Electropherograms of Dns-D,L-Asp (**a**), Dns-D,L-Ser (**b**), and Dns-D,L-Ala (**c**) on bare, PDA-coated column, and PDA/L-Arg@capillary. Buffer pH at 8. Other CE experimental conditions are the same as Figure 3.

**Figure 6 molecules-26-01800-f006:**
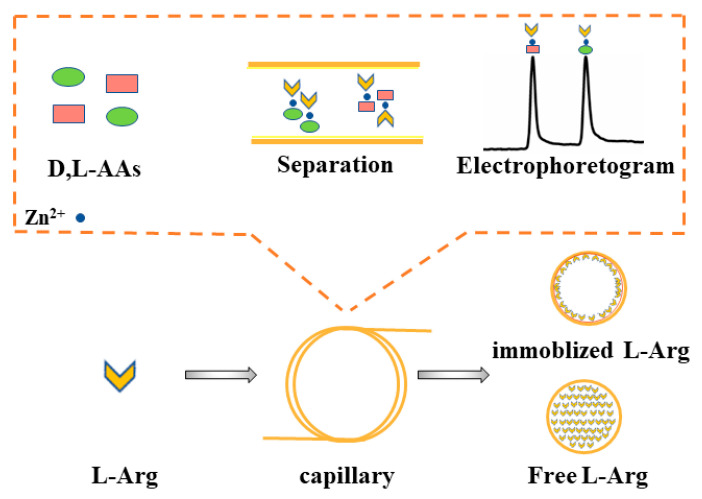
The possible mechanism of the CLE-CEC enantioseparation process.

## Data Availability

The data presented in this study are available on request from the corresponding author.

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
