# Peer review of "Polydopamine-Assisted Rapid One-Step Immobilization of L-Arginine in Capillary as Immobilized Chiral Ligands for Enantioseparation of Dansyl Amino Acids by Chiral Ligand Exchange Capillary Electrochromatography"

_molecules, 2021, doi:10.3390/molecules26061800_

Round 1

Reviewer 1 Report

I review the manuscript: “Polydopamine-assisted rapid one-step immobilization of L-arginine in capillary as immobilized chiral ligands for enantioseparation of dansyl amino acids by chiral ligand exchange capillary electrochromatography”. The manuscript idea is interesting, and I agree with the authors that it is promising technique. However, it requires additional fundamental experiments in order to support the results obtained.

Sections 2.1, 2.2 and 2.3 requires proper references for support the ideas stated. The conclusions are based on authors suppositions.

Characterization section requires a deep edition. There is not evidence of formation of polydopamine.

The amino acids were derivatized with Dansyl chloride. This reaction block the amine group which is fundamental for coordination to Zn(II) ion. There must be an additional factor involved in the separation.

Some critical points:

Please follow the instruction for authors. The sections must be: Introduction, Results, Discussion, Materials and Methods, Conclusions.

Introduction section is too long, and it is not stated the main idea of the manuscript. It is not any description about the mechanism involved for chiral separation using polydopamine-L-Arginine modified capillary.

Section 2.1. Characterization section must be reviewed carefully. There is not evidence of formation of polydopamine, it can be a dopamine adsorption. SEM micrographs quality are not enough to demonstrate the coatings (it is not clearly observed). I am quite sure that silica capillary must present Si-OH, Si-O and O-Si-O bands. In fact, the FTIR spectra of an unmodified capillary is similar than the presented in Figure 3.a. Additionally, the C=O signal assigned incorrectly, carbonyl groups vibration must be around 1600 cm-1.

Section 2.2.1. The authors describe the results but there is not any discussion. It must be included an explanation about the change of the ion mobility as the concentration increase (dopamine an L-Arg). Why in both cases the mobility changes the tendency at higher concentrations?  

Section 2.3. Why it was evaluated the Zn(II) concentration. It must be included a separation mechanism in the introduction section in order to understand the experimental methodology proposed.

The “optimized” methodology it was not included in the manuscript. I suggest including a new section with the separation methodology description. It is not clear if the separation was in normal or inverse polarity, the solutions sequence employed for analysis (composition of each one).

The authors consider all the suggestions proposed. In my opinion, it can be accepted.

Author Response

Response to Reviewer 1 Comments

Point 1: Sections 2.1, 2.2 and 2.3 requires proper references for support the ideas stated. The conclusions are based on authors suppositions.

Response 1: Thanks for the comment and suggestion! The related references for support the discussions have been supplemented in the section 2.1, 2.2 and 2.3 in the revised manuscript.

Point 2: Characterization section requires a deep edition. There is no evidence of formation of polydopamine.

Response 2: Thanks for the comment! In order to further demonstrate the successful formation of PDA and PDA/L-Arg composite, the supplementary experiments of FTIR were carried out. As shown in Figure 3a in the revised manuscript, the dopamine’s characteristic absorption peaks at 1326 cm-1 (C-H bending vibration), 1370 cm-1 (C-O-H deformation vibration) and 1239 cm-1 (C-N stretching vibration) were not discernible in the FTIR spectra of PDA and PDA/L-Arg composite, confirming that the PDA and PDA/L-Arg hybrid samples were free of dopamine and successfully synthesized. Besides, the PDA/L-Arg hybrid sample not only retained the characteristic absorption features of PDA but also exhibited the new absorption peak at 1697 cm-1, which could be ascribed to the vibration of C=O in L-Arg molecules. On the other hand, the high-resolution FESEM measurement of the inner surface of PDA/L-Arg@capillary was conducted. As shown in Figure 2B in the revised manuscript, a rough inner surface with some visible small aggregates could be observed, which also demonstrated the successful formation of L-Arg modified PDA coating on the capillary inner wall. The related supplements and discussions have been supplemented in the section 2.1.1 and 2.1.2 in the revised manuscript.

Point 3: The amino acids were derivatized with Dansyl chloride. This reaction block the amine group which is fundamental for coordination to Zn(II) ion. There must be an additional factor involved in the separation.

Response 3: Thanks for the comment! It is true that the derivatization of amino acids by dansyl chloride may have a slight effect on the formation of diastereomeric ternary metal complexes, which play a key role in the chiral separation of CLE-CE. However, the coordination atoms (e.g. carboxyl oxygen, amino nitrogen, sulfonyl oxygen, etc.) which could be complexed with Zn(II) ion still existed after the derivatization (Inorganic Chemistry 1991, 30, 1651-1655). In addition, A lot of previous studies related to the enantioseparation of D, L-AAs based on the method of CLE-CE also adopted the same strategy of dansyl derivatization to address the issues on the detection of D, L-AAs. Therefore, Dns-D, L-AAs were chosen as the model chiral analytes in this study.

Point 4: Please follow the instruction for authors. The sections must be: Introduction, Results, Discussion, Materials and Methods, Conclusions.

Response 4: Thanks for the comment! The structure of the article has been adjusted in the revised manuscript according to the instruction for authors.

Point 5: Introduction section is too long, and it is not stated the main idea of the manuscript. It is not any description about the mechanism involved for chiral separation using polydopamine-L-Arginine modified capillary.

Response 5: Thanks for the comment! The introduction section has been appropriately simplified. Moreover, the chiral separation mechanism of CLE-CEC systems has been supplemented in the introduction section and the further discussions on enantioseparation mechanisms using PDA/L-Arg@capillary were investigated and discussed in the section 2.6 in the revised manuscript.

Point 6: Section 2.1. Characterization section must be reviewed carefully. There is not evidence of formation of polydopamine, it can be a dopamine adsorption. SEM micrographs quality are not enough to demonstrate the coatings (it is not clearly observed). I am quite sure that silica capillary must present Si-OH, Si-O and O-Si-O bands. In fact, the FTIR spectra of an unmodified capillary is similar than the presented in Figure 3.a. Additionally, the C=O signal assigned incorrectly, carbonyl groups vibration must be around 1600 cm-1.

Response 6: Thanks for the comments and suggestions! In order to further demonstrate the successful formation of PDA and PDA/L-Arg composite, the supplementary experiments of FTIR were carried out. As shown in Figure 3a in the revised manuscript, the dopamine’s characteristic absorption peaks at 1326 cm-1 (C-H bending vibration), 1370 cm-1 (C-O-H deformation vibration) and 1239 cm-1 (C-N stretching vibration) were not discernible in the FTIR spectra of PDA and PDA/L-Arg composite, confirming that the PDA and PDA/L-Arg hybrid samples were free of dopamine and successfully synthesized. Besides, the PDA/L-Arg hybrid sample not only retained the characteristic absorption features of PDA but also exhibited the new absorption peak at 1697 cm-1, which could be ascribed to the vibration of C=O in L-Arg molecules.

On the other hand, according to your kind suggestion, the high-resolution FESEM measurement of the inner surface of PDA/L-Arg@capillary was also conducted. As shown in Figure 2B in the revised manuscript, a rough inner surface with some visible small aggregates could be observed, which also demonstrated the successful formation of L-Arg modified PDA coating on the capillary inner wall.

Next, ATR-FTIR measurements were employed to probe the possible PDA/L-Arg coatings on the quartz plates. The characteristic absorption of 801 cm-1 (Si–OH) and 982 cm-1 (Si–O–Si) derived from silica could be observed in unmodified plate, PDA functionalized and PDA/L-Arg functionalized plates. In comparison with unmodified plate, PDA and PDA/L-Arg coated plates exhibited clear characteristic bands associated with PDA at around 1448 cm-1, 1524 cm-1, 1610 cm-1 3221 cm-1 and 3395 cm-1, according with the characterization results of FTIR. Moreover, PDA/L-Arg coated plate also exhibited absorption feature of C=O vibration at 1692 cm-1, again confirming the successful modification of L-Arg in PDA coating (Figure 3b). Additionally, the inaccurate statement in the characteristic absorption of C=O vibration has been corrected in the revised manuscript.

Point 7: Section 2.2.1. The authors describe the results but there is not any discussion. It must be included an explanation about the change of the ion mobility as the concentration increase (dopamine an L-Arg). Why in both cases the mobility changes the tendency at higher concentrations?

Response 7: Thanks for the comments and suggestions! As shown in Figure S1b in the supplementary data, with the molar ratio of dopamine to L-Arg increased from 2:1 to 2:5, both of the resolutions and migration times of Dns-D, L-Ala gradually increased on PDA/L-Arg@capillary, which could be attributed to more L-Arg molecules existing in the hybrid coatings and increased interaction strength between PDA/L-Arg coatings and the model analyte. However, the enantioseparation efficiency began to decrease when the molar ratio of dopamine to L-Arg was further increased to 2:7, which might be because the excessively high alkalinity of the reaction solution resulted from the excessive L-Arg was not conducive to the self-polymerization of dopamine and the co-polymerization between dopamine and L-Arg. The detailed explanation about the change of the ion mobility as the concentration increase has been supplemented in the section 2.2.1 in the revised manuscript.

Point 8: Section 2.3. Why it was evaluated the Zn(II) concentration. It must be included a separation mechanism in the introduction section in order to understand the experimental methodology proposed.”

Response 8: Thanks for the comments! The chiral separation mechanism of CLE-CEC systems has been supplemented in the introduction. Specifically, the chiral recognition mechanism of CLE-CE and CLE-CEC is based on the formation of diastereomeric ternary metal complexes between the chiral ligands and the analytes. The chiral separation can be obtained owing to the different stability constants of the metal complexes. Therefore, Zn(II) ion plays an important role in the enantioseparation of CLE-CEC systems. The results shown in the section 2.3 in the revised manuscript also indicated that the concentration of Zn(II) has great impact on the CLE-CEC enantioseparation effects.

Point 9: The “optimized” methodology it was not included in the manuscript. I suggest including a new section with the separation methodology description. It is not clear if the separation was in normal or inverse polarity, the solutions sequence employed for analysis (composition of each one).

Response 9: Thanks for the comments and suggestion! The separation was in inverse polarity and the operating voltage is -20 kV. Before being measured, the modified capillaries were flushed with running buffer for 15 min, and each 3 min with distilled water and buffer solution between consecutive injections. The related experimental details have been supplemented in the section of Materials and Methods (section 3.2-3.4) in the revised manuscript.

Reviewer 2 Report

The manuscript describes the preparation and application of a new type of coated capillary for chiral capillary electrochromatography. Authors combine immobilized dopamine-L-arginine polymer on the capillary surface with separation buffer containing Zn(II) ions and L-arginine. The separation potential was successfully tested on racemic mixtures of 11 amino acids. The application of the method was demonstrated on the enzyme kinetics of L-glutamic dehydrogenase.

I consider the manuscript a nice piece of pioneer work in the field of chiral CEC with a coated capillary. The work is well structured and organized. All results are supported by a number of experimental data. However, the presented method is rather too complicated and time-consuming in comparison with the established methods for amino acids enantioseparation.

I would recommend a revision of the following parts prior to the publication in Molecules.

  1. Line 66 – A spell check required. „ …seventy percent acetonitrile in running buffer. [25]. nevertheless…“
  2. Line 71 – The consumption of solvents in CE is very low. The environmental aspect of the non-organic separation buffer is very disputable in this case. Heating the capillary for 120 minutes might be considered environmental less friendly than using few microlitres of methanol or acetonitrile.  
  3. Line 103 – Figure 1 is not well-arranged. It combines polymerization reaction with a capillary coating preparation. I would recommend providing the proper reaction scheme of dopamine with L-arginine only. A model of the capillary with the coating fits perfectly in the graphical abstract.
  4. Line 128 – Figure 2 – Why do you show two FESEM images of the inner wall of the capillary (a-b, c-d)? I would consider moving this figure into supporting material.
  5. Part 2.2.1 - Have the authors tried different ratios PDA/L-Arg? Both components play role in the polymer formation, therefore it is necessary to assess their concentrations together, not separately.
  6. Part 2.2 – How the authors controlled the final coating of the capillary? Have they performed a reproducibility study? What is the stability of the coating? How many analyses could be performed without signs of degradation?
  7. Line 261 – A spell check required. „Michaelis-Menten’s constant (Km)“ should be changed to Km.
  8. Part 4 – Separation conditions are missing the experimental part. Was the capillary somehow adjusted prior to the measurements?

Author Response

Response to Reviewer 2 Comments

Point 1: Line 66 – A spell check required. “…seventy percent acetonitrile in running buffer. [25]. nevertheless…”

Response 1: Thanks for the comment! The spelling mistake as you indicated has been corrected in the revised manuscript.

Point 2: Line 71 – The consumption of solvents in CE is very low. The environmental aspect of the non-organic separation buffer is very disputable in this case. Heating the capillary for 120 minutes might be considered environmental less friendly than using few microlitres of methanol or acetonitrile.

Response 2: Thanks for the comment! It is true that statements of “…the usage of environmentally hazardous organic solvents in buffer solution…” is inappropriate and disputable. We have corrected as “…the overuse of organic solvents in buffer solution which has negative effects on biological samples analysis.”

Point 3: Line 103 – Figure 1 is not well-arranged. It combines polymerization reaction with a capillary coating preparation. I would recommend providing the proper reaction scheme of dopamine with L-arginine only. A model of the capillary with the coating fits perfectly in the graphical abstract.

Response 3: Thanks for your kindly suggestions. the Figure 1 in the original manuscript is really not well-arranged. Figure 1 has been adjusted to the schematic of the hydrothermal-assisted preparation of PDA/L-Arg@capillary in the revised manuscript.

Point 4: Line 128 – Figure 2 – Why do you show two FESEM images of the inner wall of the capillary (a-b, c-d)? I would consider moving this figure into supporting material. 

Response 4: Thanks for the comment! There is somewhat misunderstanding. The two FESEM images in the original manuscript have different magnification. In order to more clearly characterize the surface morphology of the capillary inner wall, the high-resolution FESEM measurement of PDA/L-Arg@capillary was further conducted and the updated Figure 2 has been supplemented in the revised manuscript.

Point 5: Part 2.2.1 - Have the authors tried different ratios PDA/L-Arg? Both components play role in the polymer formation, therefore it is necessary to assess their concentrations together, not separately.

Response 5: Thanks for the comments and suggestions! It is indeed necessary to assess their concentrations together. The effects of the molar ratio of dopamine and L-Arg on enantioseparation capacity have been supplemented in section 2.2.1 in the revised manuscript according to your kind suggestion.

Point 6: Part 2.2 – How the authors controlled the final coating of the capillary? Have they performed a reproducibility study? What is the stability of the coating? How many analyses could be performed without signs of degradation?

Response 6: Thanks for the comment! Under the optimal CLE-CEC operation conditions, the repeatability and stability of PDA/L-Arg@capillary was evaluated based on the RSDs of retention time of Dns-D, L-AAs. The RSDs of intra-day, inter-day and column-to-column were all below 5.8% (Table S3). Besides, PDA/L-Arg@capillary could be performed for 80 consecutive injections without significant signs of degradation in the retention times and the enantioseparation performance (Figure S5). The results confirmed that PDA/L-Arg@capillary had good repeatability and stability. The related results and discussion have been supplemented in section 2.5 in the revised manuscript and revised supplementary data (Table S3, Figure S5).

Point 7: Line 261 – A spell check required. “Michaelis-Menten’s constant (Km)” should be changed to Km.

Response 7: Thanks for the comment! The spelling error has been corrected in the revised manuscript.

Point 8: Part 4 – Separation conditions are missing the experimental part. Was the capillary somehow adjusted prior to the measurements?

Response 8: Thanks for the comment! Before being measured, the modified capillaries were flushed with running buffer for 15 min, and each 3 min with distilled water and buffer solution between consecutive injections. The separation conditions have been supplemented in the section 3.4 in the revised manuscript.

Round 2

Reviewer 1 Report

The authors considered all the suggestions proposed. The quality of the manuscript has been improved. In my opinion, it can be accepted.